# Stability of Porous Polymeric Membranes in Amine Solvents for Membrane Contactor Applications

**DOI:** 10.3390/membranes13060544

**Published:** 2023-05-23

**Authors:** Denis Kalmykov, Sergey Shirokikh, Evgenia A. Grushevenko, Sergey A. Legkov, Galina N. Bondarenko, Tatyana S. Anokhina, Sergey Molchanov, Stepan D. Bazhenov

**Affiliations:** A.V. Topchiev Institute of Petrochemical Synthesis RAS, 29 Leninsky Prospekt, 119991 Moscow, Russia

**Keywords:** membrane stability, carbon dioxide, membrane contactor, oxygen removal, PP, PVDF, PTFE, PES, PA

## Abstract

Membrane gas–liquid contactors have great potential to meet the challenges of amine CO_2_ capture. In this case, the most effective approach is the use of composite membranes. However, to obtain these, it is necessary to take into account the chemical and morphological resistance of membrane supports to long-term exposure to amine absorbents and their oxidative degradation products. In this work, we studied the chemical and morphological stability of a number of commercial porous polymeric membranes exposed to various types of alkanolamines with the addition of heat-stable salt anions as a model of real industrial CO_2_ amine solvents. The results of the physicochemical analysis of the chemical and morphological stability of porous polymer membranes after exposure to alkanolamines, their oxidative degradation products, and oxygen scavengers were presented. According to the results of studies by FTIR spectroscopy and AFM, a significant destruction of porous membranes based on polypropylene (PP), polyvinylidenefluoride (PVDF), polyethersulfone (PES) and polyamide (nylon, PA) was revealed. At the same time, the polytetrafluoroethylene (PTFE) membranes had relatively high stability. On the basis of these results, composite membranes with porous supports that are stable in amine solvents can be successfully obtained to create liquid–liquid and gas–liquid membrane contactors for membrane deoxygenation.

## 1. Introduction

According to global statistics, CO_2_ emissions continue to be the main cause of climate change. The desire to limit the impact of these emissions led to the signing of the Paris Agreement [1], a program document that significantly affects the tax and customs policy of many states [2,3]. In addition, carbon dioxide may be of interest for direct use in the development of gas [4] and oil [5] fields. Currently, the most technologically advanced and mature method of CO_2_ capture is amine purification, an absorption process using various alkanolamines [6]. This technology can be more successfully implemented using membrane contactors due to their modularity, compactness and large specific mass transfer surface area [7].

Membrane contactors can be used for various aspects of amine absorption technology: (a)Direct CO_2_ absorption from gas mixtures using aqueous solutions of amines: The overwhelming majority of work in this area is devoted to precisely this stage of the process [7,8];(b)Membrane CO_2_ desorption/stripping: In the last decade, this direction of membrane contactor application has been intensively studied [9,10], and in this case, membrane contactors are used to remove CO_2_ from amines at elevated temperatures;(c)Membrane deoxygenation of amine solvents: This is a relatively new field of the application of gas–liquid or liquid–liquid membrane contactors. The promise of this approach was first demonstrated by TNO using the Dissolved Oxygen Removal Apparatus (DORA) at TRL6 [11]. The relevance of this problem is due to the fact that oxygen is often present in the mixture being purified, which leads to the oxidative degradation of amines [12,13,14]. In addition to the harm caused by the direct intensification of corrosion in amine solvents, degradation, and the oxidation of amines [15,16,17,18,19,20,21,22,23], it is important to note the effect of oxidation products on the overall performance of the system. Thus, under the influence of oxidation products (carboxylic acids, amides, aldehydes, and amino acids), the physicochemical degradation of the amine occurs by the foaming, erosion, and precipitation of non-regenerated heat-stable salts (HSSs) that accumulate in the system and lead to general pollution. In addition, these compounds are direct corrosion catalysts [24,25], suggesting the autocatalytic degradation of the absorbent.

Both porous membranes made of hydrophobic polymers and composite membranes with thin permeable layers can be used in membrane contactors. The use of composite membranes prevents the penetration and wetting of the porous membrane structure by the absorbent. However, the selective layers of the composite membranes are permeable to solvent vapors [26,27]. Amine vapors can be adsorbed/condensed in the porous supports and affect their structure. 

In all the above cases (a–c), the membranes used in the contactors are exposed to amines at elevated temperatures (60–100 °C), often with the presence of oxidative degradation products [28]. Accordingly, both in the case of porous and composite membranes, it is important to study the stability of the used membranes or supports under the long-term effect of degraded amine solvents. 

The stability of various polymeric membranes is being studied by researchers for many applications. For example, in a number of works [29,30,31,32], the stability of polypropylene (PP), polyvinylidenefluoride (PVDF), polytetrafluoroethylene (PTFE) and polyethylene (PE) membranes was studied. In particular, the authors of [29] studied the stability of polypropylene membranes in NaOH during the separation of H_2_S and CO_2_. On the other side, despite the alkaline environment, even after six months of exposure, the membranes almost did not degrade. In [30], the authors studied the effect of several organic solvents (n-hexane, ethyl alcohol, p-xylene, monochlorobenzene, dimethylformamide, and dimethylsulfoxide) on the properties of microporous hollow polyethylene fiber membranes after contact for 30 min at room temperature. After treatment with various solvents, the pore size and membrane permeability increased. At the same time, this effect was more pronounced for solvents with high surface tension.

Within the framework of works on amine CO_2_ solvents, the works devoted to the analysis of membrane stability in amine solvents are of significant interest. In particular, the stability of the hollow PP fiber membranes Celgard X40-200 and Celgard X50-215 in diethanolamine (DEA) was considered in [33]. Furthermore, separate studies were carried out in amine solvents with a high content of CO_2_. The results showed a decrease in membrane degradation in solutions with a high content of CO_2_. However, in this case, there was also a decrease in the performance of the membrane contactor. According to [34], the degradation of PP membranes increased as the temperature rose to 60 °C. A number of works have been devoted to studying the stability of various membrane materials in membrane contactors in contact with amine solvents [35]. In this context, the stability of membranes made of various polymeric materials was studied: PP, PVDF [36,37,38], low-density polyethylene (LDPE) [39,40], and PTFE [41,42]. In particular, the change in a number of parameters of PVDF membranes during long-term (15 days) exposure to various amine media was studied. In [43,44,45], studies on the stability of hybrid and composite membranes with various additives were also carried out. The introduced additives could increase the resistance of membranes to degradation. It should be especially noted that, in all of these works, the authors separately noted the influence of the chemical interaction of the support material and amine solvents.

Since aqueous solutions of alkanolamines containing oxidative degradation products are used for CO_2_ absorption, it may be of interest to study changes in the structure of membrane materials after their contact with solutions of inorganic salts. The authors of [31] compared the surface morphology of Celgard 2500 and Accurel 1E-PP polypropylene membranes after their exposure to water and 30 wt.% CaCl_2_ solution for 72 h. It was shown that the porosity and pore size of membranes increased after 72 h of interaction with water due to the penetration of a non-wetting liquid into the pores with their subsequent expansion. On the other hand, in the case of using a CaCl_2_ solution, no significant changes in the morphology of the membrane surface were observed. This could be associated with a less intense penetration of the CaCl_2_ solution due to a higher surface tension [31]. Similarly, the authors of [32] studied the stability of hollow PTFE and PVDF fibers when used for the separation of olefin/paraffin. These hollow fibers were in contact with a 0.2 M silver nitrate solution for 2 months. It was demonstrated that the characteristics of the PTFE did not change over time. On the other hand, the transport characteristics of hollow PVDF fibers decreased linearly throughout the entire study period due to the better wettability of PVDF by the solvent and the accumulation of a significant amount of silver on the membrane surface, which changed the surface morphology.

Additionally, one of the solutions to the problem of the oxidative degradation of amine absorbents is the introduction of oxygen scavengers, such as sodium/potassium sulfite or bisulfite, into the solvent composition [46,47]. Such oxygen scavengers allow the oxygen content in the solvents to be reduced to levels of several ppm. However, these compounds are toxic to humans, hazardous to the environment, and can lead to foaming solutions. In addition, oxygen scavengers can potentially also have a negative effect on the membrane material if they are present in alkanolamine solutions in membrane contactors.

Accordingly, the effect of amine solvents containing oxidative degradation products, as well as oxygen scavengers, on the structure and morphology of various polymeric membranes is of interest. This issue is especially interesting for their use as composite membrane supports in membrane contactors operating in an absorbent medium. Thus, within the framework of this work, it was proposed to study the stability of a number of commercial porous polymer membranes (potential membrane supports for composite membranes) in various types of alkanolamines at elevated temperatures with the addition of HSS ions, which simulates the composition of real industrial solutions. The study of membrane stability at elevated temperatures is due to their potential use in membrane contactors. During amine purification from CO_2_, the maximum temperature of the carbon-dioxide-filled solvent can be up to 100–120 °C (the temperature of regeneration of absorbents). Membrane contactors can also be used under these conditions to implement such a process [48,49]. An analysis of the resistance of membranes to the interaction with a model solution of oxygen scavengers was also carried out. 

## 2. Materials and Methods

### 2.1. Materials and Reagents

#### 2.1.1. Membranes

In this work, five commercial porous polymer membranes (commonly used as porous supports for composite membranes) were studied: fluoropolymer, polyolefin, polyethersulfone, and polyamide membranes. Some information about the investigated membranes can be found in Table 1.

We used commercial flat membranes made of polypropylene (PolySep™, GE Osmonics Labstore, Minnetonka, MN, USA), polyvinylidenefluoride PVDF-022, polytetrafluoroethylene MFF4-020, polyethersulfone PES-020, and polyamide MMK-010 (Technofilter RME, Vladimir, Russia).

#### 2.1.2. Amine Solutions

Within the framework of this work, four industrially used amines were studied. In addition to the model solutions, a real industrial degraded absorbent was also applied. Table 2 lists the amine solutions used.

Monoethanolamine (MEA, LLC TD HIMMED (Moscow, Russia)), N-methyldiethanolamine (MDEA) manufactured by GC Sintez OKA (Dzerzhinsk, Russia), 2-Amino-2-methylpropane-1-ol (AMP) and piperazine (PZ) manufactured by Sigma-Aldrich (Saint Louis, MI, USA) were used for the preparation of the model amine solvents. An aqueous solution of Na_2_SO_3_ (50 g/L) was used with the addition of a catalyst Co(NO_3_)_2_ (0.05 g/L, LLC TD HIMMED (Moscow, Russia)) as a model oxygen scavenger.

#### 2.1.3. Heat-Stable Salt Anions

The following anions in the form of the corresponding acids were added to the model solutions of MEA, MDEA, AMP, and PZ to simulate the presence of absorbent degradation products (HSSs). Acid concentrations were taken on the basis of the data presented in [50,51] (Table 3).

The solution was prepared using distilled water (TIPS RAS, Moscow, Russia) and a set of acids: formic acid, oxalic acid, acetic acid, nitric acid, sulfuric acid, and hydrochloric acid (LLC TD HIMMED, Moscow, Russia). 

### 2.2. Membranes Characterization

#### 2.2.1. Long-Term Treatment of Porous Polymer Membranes with Amine Solvents

To assess resistance to long-term exposure to various compounds, samples of porous polymeric membranes (~2 × 3 cm) were placed in different liquids (model degraded-amine CO_2_ solvents MEA, MDEA, AMP, PZ containing HSS anions, industrial DEA solution, and a solution of a model oxygen scavenger Na_2_SO_3_). The closed containers with solutions and membranes were placed in an oven (Binder, Tuttlingen, Germany) at 100 °C and kept for at least 14 days. After that, membranes were removed from the solutions, kept at room temperature in ethanol (2 h) and water (2 h) to remove residual absorbents, air-dried, and examined by a number of physicochemical methods of analysis. It should be noted that all the applied methods of analysis were also used to study the properties of the initial membranes before exposure to amine solvents.

#### 2.2.2. FTIR Spectroscopy

The possible changes in the chemical structure of polymeric membranes that were not destroyed after exposure to selected liquids were studied by Fourier transform infrared (FTIR) spectroscopy. The FTIR spectra of the membranes were recorded by the reflection method on an IFS-66v/s IR Fourier spectrometer (Bruker, Billerica, MA, USA) using an attenuated total reflection (ATR) attachment (ZnSe crystal, scan 30, resolution 2 cm^−1^, range 600–4000 cm^−1^). Mathematical processing of the spectra (reduction to baselines, normalization, etc.) was carried out using the OPUS-7.0 software package (Bruker, Billerica, MA, USA). The wavenumber (ν, cm^−1^) was set as the abscissa axis of all the presented spectra, and the dimensionless intensity value, optical density (D), was set as the ordinate axis.

#### 2.2.3. Atomic Force Microscopy

Membrane samples were also studied by atomic force microscopy (AFM). AFM was used to quantitatively analyze the morphology of the selective layer of the hollow fiber. A Horiba Smart SPM scanning probe microscope (Vénissieux, Lyon, France) was used with the semi-contact mode of atomic force microscopy. We used probes from Nanoandmore (Wetzlar, Germany) with parameters F = 330 kHz, C = 42 N/m, L = 125 µm, and R ≤ 10 nm. 

The porosity and surface roughness of the test samples were calculated using the Gwyddion software (version 2.62, Czech Institute of Metrology, Brno, Czech Republic) from their AFM images. 

To calculate the porosity of the membrane samples, the area of the AFM images was analyzed using the Gwyddion software (version 2.62). The porosity of the sample was determined as a percentage of the area of the pores to the total area of the image.

The roughness was determined based on the automatically determined depth of dots on the image by two parameters: the average roughness *R_a_* (Formula (2)) and the root-mean-square roughness *R_q_* (Formula (2)).
(1)$$R_{a}=\frac{1}{l_{r}}\int_{0}^{l_{r}}|z(x)|dx$$
(2)$$R_{q}=\sqrt{\frac{1}{l_{r}}\int_{0}^{l_{r}}z{\left(x\right)}^{2}dx}$$
where *l_r_* is the length of the baseline and *z*(*x*) is the deviation from the baseline. 

#### 2.2.4. Pore Size Measurements

The pore size was measured by liquid–liquid displacement porosimetry using the porometer POROLIQ 1000 ML (Porometer, Belgium). Membrane pore size analysis was performed by the liquid–liquid displacement method using water-saturated isobutanol and isobutanol-saturated water as a solvent pair. The size was calculated according to the procedure described in detail in [52]. The porous structure was characterized by the diameter of the largest pore (d_max_) and the diameter of the smallest pore (d_min_), as well as the mean flow pore size d_MFP_. The mean flow pore size value is defined as the pore size at which 50% of the flux penetrates through the larger pores and 50% of the flux penetrates through the smaller pores of the membrane.

#### 2.2.5. Gas Permeance Measurements

To study changes in the transport and separation characteristics of porous polymeric membranes, gas permeance was measured according to the volumetric method described in [53] using the individual gases nitrogen, oxygen, and carbon dioxide (MGPZ, Moscow, Russia).

## 3. Results and Discussion

### 3.1. Changes in the Chemical Structure of Polymeric Membranes

To analyze changes in the chemical structure of the polymeric materials of the studied membranes under the influence of amine solvents, an analysis was carried out by FTIR spectroscopy. For example, Figure 1 shows the FTIR spectra of PVDF and PTFE membrane samples before and after treatment with various amine solvents and Na_2_SO_3_ solution. The FTIR spectra of PP, PES, and PA membrane samples are found in Appendix A, respectively. It should be noted that some of the membrane samples were destroyed to a state in which their study seemed difficult. Accordingly, the data on such samples are not presented in the relevant sections.

In the FTIR spectrum of the PVDF membranes, the presence of ammonium cations and carbamate anions is manifested in almost all the samples, with many of them on the surface of the PVDF membrane after exposure to AMP (PVDF–AMP) and MEA (PVDF–MEA). In the case of the PVDF–MEA sample, the formed salts completely covered the membrane surface in such a way that no bands of the polymeric material of the membrane (PVDF) appear in the spectrum. Therefore, even very intense bands of C-F bonds in the region of 1200 cm^−1^ from PVDF do not appear. The technique for recording spectra using a ZnSe crystal makes it possible to record the spectrum of a membrane surface with a thickness of up to 1.4 μm. Consequently, it can be concluded that HSSs covered the PVDF–MEA sample with a layer no less than 1.4 µm-thick. In the FTIR spectrum of the PVDF-AMP sample, the bands from HSSs (1500–1700 cm^−1^) also have a high intensity, but at the same time, all the characteristic bands corresponding to PVDF also appear. In the spectra of PVDF–DEA, PVDF–AMP–MDEA, PVDF–Na_2_SO_3_, the HSS intensity of the bands is 10 times lower, and in the PVDF–Na_2_SO_3_ sample, the salt content does not exceed 0.05 mol.%.

In the FTIR spectra of the PTFE samples (Figure 1b), there are almost no impurity bands corresponding to HSSs. In the PTFE–DEA and PTFE–MEA samples, only very weak bands of C-O bonds at 1023 cm^−1^, ammonium cations at 1620 cm^−1^, and C-H bonds in the region of 2840–2960 cm^−1^ appear. However, in the PTFE–DEA sample, the content of these impurities does not exceed 0.02 mol.%, and in the PTFE–MEA sample, their content is approximately 3 times less.

The FTIR spectra of the PP samples (Appendix A) show weak signs of the presence of solvent degradation products. According to the content of HSSs on the surface of PP membranes, the following series can be built: MEA > DEA > Na_2_SO_3_ > MDEA > AMP > PZ. At the same time, in the PP–MEA sample, the salt content does not exceed 0.1 mol.% relative to PP.

In the PES FTIR spectra (Appendix A), there are no characteristic peaks corresponding to the alkyl (CH_2_ and the CH_3_) groups or the stretching vibrations of C−H bonds in the region of 2840–2960 cm^−1^. However, the spectra of all the samples aged in amines contain weak bands in this region. At the same time, in the case of the PES–MDEA sample, the characteristic peaks of MDEA are more pronounced. Thus, amines were present in small amounts (less than 0.02 mol.%) on the polymer surface.

In the FTIR spectrum of the original PA membrane (Appendix A), there are pronounced signs of the presence of water on its surface. Moreover, water molecules are associated with N−C=O amide groups in the composition of the polymer, since relatively wide bands are observed that are characteristic of water molecules (3405 and 3170 cm^−1^), shifted to the region of long waves. At the same time, the bands corresponding to the C=O bond in the PA spectrum are also shifted to the long-wavelength region: 1633 cm^−1^ (amide I) and 1538 cm^−1^ (amide II). After the amine solvents, water practically does not appear on the surface of the samples. It is almost impossible to identify salts of ammonium cations and carbamate ions in the region of 1700–1550 cm^−1^ on the FTIR spectra due to very intense bands of the PA amide group in this region. However, in all the studied spectra of PA membranes exposed to amines, new bands appear in the region of 1100–950 cm^−1^, which may correspond to C−OH bonds. The spectrum of the PA–Na_2_SO_3_ sample shows the presence of the most intense bands in this region (peak at 1016 cm^−1^), which are not characteristic of sulfite ions (in the region of 1100 cm^−1^). This may be due to the more intensive processes of oxidation of amide units in Na_2_SO_3_ with the formation of C−OH groups.

### 3.2. Changes in the Morphology of Polymeric Membranes

Changes in the morphology of the porous polymeric membranes as a result of exposure to model solutions of amine solvents were studied by the AFM method. For example, Figure 2 and Figure 3 show AFM images of PVDF and PES membrane samples before and after treatment with MEA solvents containing HSS anions to compare the various morphological changes found. A set of AFM images of membrane samples from the studied polymeric materials is presented in the Appendix A.

Table 4 presents the data obtained from AFM images of the morphological characteristics of all membrane samples before and after treatment with various amine solvents and Na_2_SO_3_ solution. Statistical quantities (minimum and maximum heights of points on the image), porosity (expressed as a percentage of the area occupied by pores relative to the total surface area), and roughness (defined as root-mean-square *R_q_* and average *R_a_* roughness) were chosen as the parameters for analysis. 

The PVDF membrane samples underwent morphological changes in all the studied liquids (Appendix A). Despite the high chemical resistance, the surface morphology of the PTFE membranes underwent certain changes in the amine solvent environment. Thus, contact with liquids led to the ‘etching’ of the surface with an increase in its roughness, which is noticeable on the large-scale AFM images (Appendix A). Apparently, the mechanical destruction of biaxially oriented fibrils occurs with the removal of a portion of the undeformed spherical nodular particles of PTFE membranes. 

As described in Section 3.1, in the studied samples of PP membranes, there were signs of a slight presence of the solvents’ chemical degradation products. At the same time, the AFM data (Appendix A) indicated a significant change in the surface morphology of all the studied PP membrane samples.

The morphology of the PES membranes also underwent changes under the influence of the model solutions used. Contact of the PES membrane with the solvents led to an increase in the surface porosity. At the same time, in the environment of primary alkanolamines (MEA, AMP), there was also a narrowing of the initial pores, which is illustrated by the images in Figure 3 and Appendix A.

The PA membrane samples were morphologically unstable in the solvents used. Contact with amines apparently led to partial ‘etching’ of the surface with an increase in roughness and the adsorption of amine molecules on the membrane surface, which is most noticeable for the samples of PA–MDEA (Appendix A). 

Table 5 summarizes the results of the analysis of chemical and morphological stability (according to FTIR spectroscopy and AFM) of porous polymeric membranes in model degraded solutions of alkanolamines, a real industrial sample of a degraded amine solvent, and a model solution of an oxygen scavenger.

According to the results obtained, the absence of chemical and morphological stability of the samples of porous polymeric membranes from PP, PVDF, PES and PA (nylon) in the environment of model degraded absorbents based on alkanolamines at elevated temperature was revealed. The conclusions drawn on the stability of the studied membranes are in good agreement with the literature data. The authors of [33] noted the morphological instability of hollow PP fibers upon contact with a 30% DEA solution for 30 days. In addition, PP and PVDF membranes demonstrated low morphological resistance to the action of amine solvents during the membrane absorption process as well [36]. Flux and CO_2_ capacity were significantly reduced within 30 days of hollow-fiber operation in a membrane absorption process using a 1M MEA solution as an absorbent. At the same time, PVDF membranes turned out to be less stable than PP ones [36]. It is important to note that changes in PVDF membrane properties are associated with changes in the surface morphology while maintaining chemical stability, even after long-term storage in amine solvents at room temperature [37]. However, as shown in our studies, an increase in temperature led to both morphological and chemical changes in the PVDF membranes.

Moreover, the PTFE membranes demonstrated a sufficiently high chemical stability with certain changes in the surface morphology, which is also consistent with the results of [41,42]. The authors of [41] demonstrated the greater stability of PTFE and PP membranes compared to PVDF membranes. The PTFE and PP membranes demonstrated a larger water contact angle than PVDF membranes at room temperature. The PTFE membranes showed greater stability after the treatment of 1-butyl-3-methylimidazolium tricyanomethanide at 100 °C for 6 h [41].

Accordingly, the results obtained in the field regarding changes in the morphology of the membranes can be associated with different levels of wettability and penetration into the pores of the solvents of various compositions and viscosities [35,36,54,55,56]. This assumption is also consistent with the literature data. In [54], the membrane absorption of CO_2_ by MEA was performed using PTFE and PVDF membranes. The PTFE membranes were able to maintain their performance after 60 h of process, while the PVDF membranes were degraded. The increase in resistance to mass transfer in membranes increased due to wetting [55]. Changes in membrane properties due to different levels of wettability have been shown not only in different materials, but also in different amines. The effect of MEA on the stability of PP membranes was comparable to that of PZ [55]. In turn, the greater MEA wettability of PVDF membranes compared to AMP and DEA led to a greater effect on membrane stability [56].

To study the effect of membrane wettability on the change in morphology, we also measured the water contact angles before and after exposure to the model solvent (Table 6). Contact angles were measured using a conventional sessile drop technique using an LK-1 goniometer (RPC OpenScience Ltd., Moscow region, Krasnogorsk city, Russia) at room temperature (23 ± 2 °C).

The data obtained are consistent with the data in Table 5 for PES and PA membranes. The change in the structure of these membranes can be mainly associated with the wettability of their surface by model solvents. The change in the structure of the PVDF membranes can also be explained by this phenomenon. However, the contact angle of the PP and PTFE membranes did not change significantly. Thus, the results obtained in the field regarding changes in the morphology and chemical structure of the surface of the PTFE and PP membranes can also be associated with the chemical interaction and adsorption of the solvent and HSS molecules.

### 3.3. Changes in Pore Size, Transport, and Separating Properties of Polymeric Membranes

The membrane samples were investigated in more detail using liquid–liquid displacement porosimetry to establish changes in the porous structure. Table 7 shows the results of the pore size measurements of the PTFE membranes. The pore size measurement results of all the studied membranes are presented in Appendix A.

The porous structure of the PTFE membranes underwent the greatest changes in the d_min_ pore size area; d_MFP_ remained practically unchanged for all the amine solvents used. This phenomenon may confirm the assumption that contact with liquids leads to the ‘etching’ of the surface, which was inferred based on AFM images. 

The PTFE membranes were further investigated in terms of their gas transport characteristics. The absence or presence of performance changes can additionally confirm the results of the study of the stability of porous polymer membranes in amine solvents. Changes in membrane transport and membrane separation characteristics were evaluated in terms of the permeance of individual gases N_2_, O_2_, and CO_2_, as well as the calculated values of the ideal selectivities for CO_2_/N_2_ and CO_2_/O_2_ gas pairs. Figure 4 and Figure 5 show the results of the nitrogen permeance measurements and the values of ideal CO_2_/N_2_ gas pair selectivities of the studied membranes, respectively. The permeance values of all membranes studied for N_2_, O_2_, and CO_2_ and ideal selectivities for CO_2_/N_2_ and CO_2_/O_2_ gas pairs are presented in the Appendix A, respectively. 

The membrane permeance measurements were consistent with the changes in their chemical and morphological structure. The average porosity and pore size mainly determine the gas permeance of the membranes. Thus, in the case of the PTFE membranes, the gas permeance changed slightly. This phenomenon is a result of the minor variation of transport porosity (d_MFP_ remained practically unchanged after exposure to amine solvents; Table 7). It should be noted that the surface porosity presented in Table 4 varies more significantly (20–60%). From our point of view, this seemingly unexpected result is due to the fact that surface porosity may be not directly related to the transport porosity of membranes. As stated above, the mechanical destruction of biaxially oriented PTFE fibrils resulted in surface ‘etching’ and significant changes in surface porosity, but mild changes in gas transport properties. On the other hand, a slight decrease in gas permeance with a noticeable decrease in porosity can also be associated with an increase in d_min_. These results also provide additional evidence for the resistance of porous polymeric PTFE membranes to amine solvents. 

At the same time, for other membranes, a similar relationship was observed between changes in surface/transport porosity and transport properties. For example, in the case of the PVDF membranes, despite a slight decrease in surface porosity (according to AFM data), nitrogen permeance significantly decreased from 430 to 340 m^3^(STP)∙(m^2^∙h∙bar)^−1^ (Appendix A), which may be due to a decrease in d_MFP_ from 0.70 ± 0.01 to 0.56 ± 0.01 μm (Appendix A) after exposure to the solvents.

## 4. Conclusions

Within the framework of this work, we analyzed the chemical and morphological stability of porous polymeric membranes during long-term exposure at elevated temperatures to model amine solvents containing degradation products, a real industrial amine solvent, and a model solution of the oxygen scavenger (Na_2_SO_3_). Commercial membranes made from fluoropolymers (PVDF, PTFE), polypropylene (PP), polyethersulfone (PES) and polyamide (PA, nylon) were studied by FTIR spectroscopy, atomic force microscopy and liquid–liquid porosimetry to establish changes in the chemical structure and morphology of the membranes. This study showed that the commercial porous polymeric membranes presented are not sufficiently chemically and morphologically stable in amine CO_2_ solvents containing HSS anion media. The porous polymeric PTFE membranes turned out to be the most stable, which makes them promising for use as porous supports for composite membranes. The results of the measurement of the pore size and the permeance of these membranes for N_2_, CO_2_, and O_2_ are consistent with changes in their chemical and morphological structure, and indicate a resistance to amine solvents. Based on the results obtained, liquid–liquid and gas–liquid membrane contactors with amine solvent polymeric membranes that are stable in CO_2_ can also be successfully created for use in membrane deoxygenation.

## Figures and Tables

**Figure 1 membranes-13-00544-f001:**
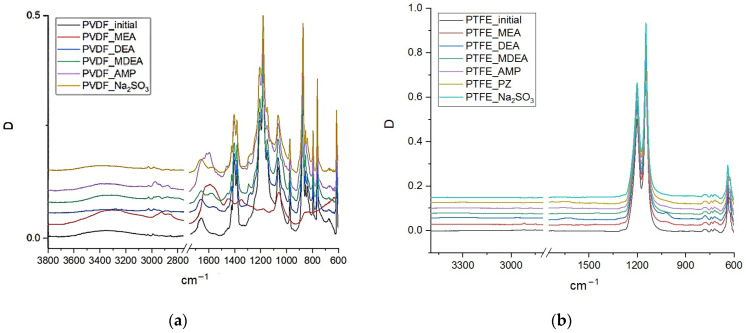
FTIR spectra of PVDF (**a**) and PTFE (**b**) membrane samples before and after exposure to model solutions.

**Figure 2 membranes-13-00544-f002:**
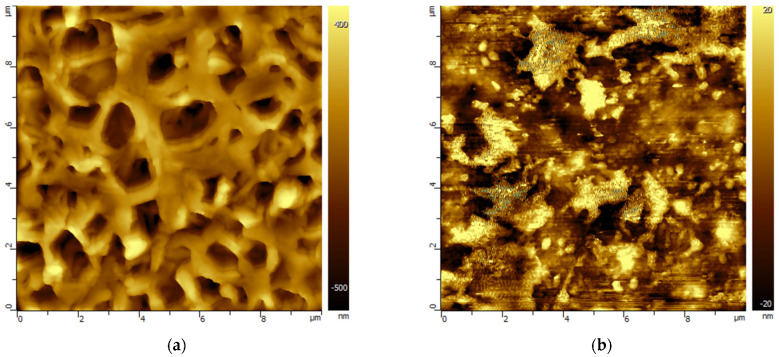
AFM images of the surface of the PVDF membrane before (**a**) and after MEA treatment (**b**). The scale of the images is 10 × 10 µm.

**Figure 3 membranes-13-00544-f003:**
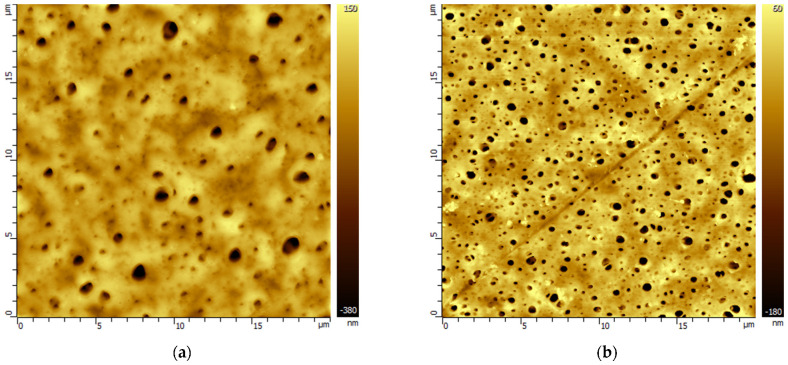
AFM images of the PES membrane surface (**a**) before and after MEA treatment (**b**). The scale of the images is 20 × 20 µm.

**Figure 4 membranes-13-00544-f004:**
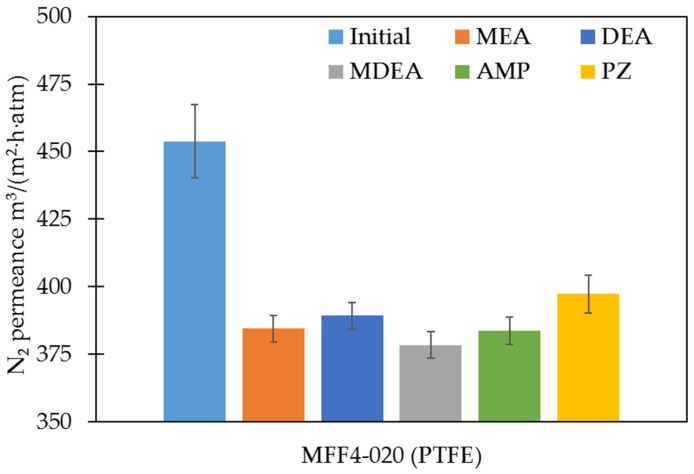
The permeance of the PTFE membranes before and after exposure to amine solutions for N_2_.

**Figure 5 membranes-13-00544-f005:**
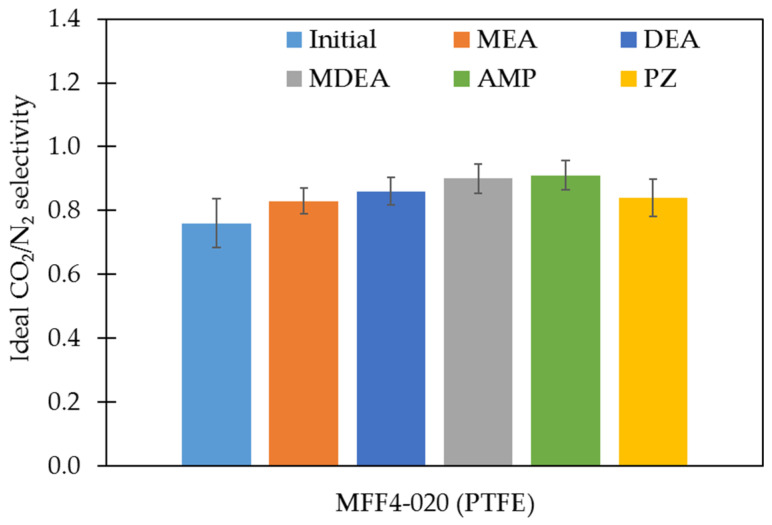
The ideal selectivity of the PTFE membranes before and after exposure to amine solutions for CO_2_/N_2_ gas pair.

**Table 1 membranes-13-00544-t001:** Porous polymeric membranes.

Material	Designation	Thickness, μm	Trading Name, Company
Fluoropolymers
Polyvinylidenefluoride	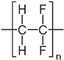	PVDF	123 ± 2	PVDF-022, Technofilter RME, Vladimir, Russia
Polytetrafluoroethylene	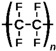	PTFE	45 ± 2	MFF4-020, Technofilter RME, Vladimir, Russia
Polyolefin
Polypropylene	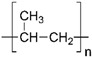	PP	95 ± 2	PolySep™, GE Osmonics Labstore, Minnetonka, MN, USA
Polyethersulfone
Polyethersulfone	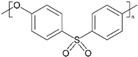	PES	79 ± 1	PES-020, Technofilter RME, Vladimir, Russia
Polyamide
Polyamide (nylon)	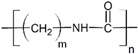	PA	111 ± 4	MCM-010, Technofilter RME, Vladimir, Russia

**Table 2 membranes-13-00544-t002:** Samples of model and real CO_2_ amine solvents.

Amine	Abbreviation	Amine Structure	Concentration, wt.%.	Comments
Monoethanolamine	MEA	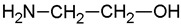	30	Solution in distilled water
N-methyldiethanolamine	MDEA	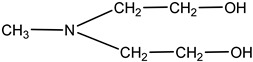	40	Solution in distilled water
2-Amino-2-methylpropan-1-ol	AMP	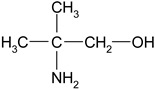	30	Solution in distilled water
Piperazine	PZ	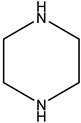	40	Solution in distilled water
Diethanolamine	DEA	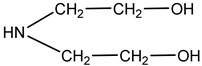	~29	A sample of a real industrial degraded solvent. The content of degradation products (heat-stable salts) is ~8000 ppm

**Table 3 membranes-13-00544-t003:** The content of HSS anions in model degraded solutions.

HSS Anion	Concentration	
	mg/L	mmol-equiv/L
Formate	1200	26.65
Oxalate	500	11.36
Acetate	50	0.85
Nitrate	200	3.23
Sulfate	400	8.33
Chloride	10	0.28
Total content of HSS anions	2360	50.7

**Table 4 membranes-13-00544-t004:** Surface parameters of the membranes studied provided by AFM.

Solution	Statistical Quantities			Porosity			Roughness	
	Minimum, nm	Maximum, nm	Δ, nm	Total Area, µm^2^	Pore Area, µm^2^	Porosity, %	Square Roughness *R_q_*, nm	Average Roughness *R_a_*, nm
PVDF-022 (PVDF)								
Initial	−400	300	700	403.0 ± 5.0	42.7 ± 3.0	11 ± 1	149.7 ± 5.0	119.6 ± 5.0
MEA	−190	190	380	402.7 ± 5.0	46.5 ± 3.0	12 ± 1	66.3 ± 3.5	52.1 ± 3.5
DEA	−500	400	900	401.0 ± 5.0	40.6 ± 3.0	10 ± 1	182.1 ± 5.0	148.3 ± 5.0
MDEA	−400	400	800	400.0 ± 5.0	32.4 ± 3.0	8 ± 1	167.2 ± 5.0	136.4 ± 5.0
AMP	−500	500	1000	401.0 ± 5.0	92.0 ± 3.0	23 ± 1	244.8 ± 5.0	203.6 ± 5.0
PZ *	−	-	-	-	-	-	-	-
Na_2_SO_3_	−1700	1400	3100	404.0 ± 5.0	53.3 ± 3.0	13 ± 1	708.2 ± 7.0	568.2 ± 7.0
MFF4-020 (PTFE)								
Initial	−360	200	560	400.0 ± 5.0	47.2 ± 3.0	12 ± 1	116.3 ± 5.0	92.8 ± 5.0
MEA	−400	300	700	398.9 ± 5.0	39.2 ± 3.0	10 ± 1	143.7 ± 5.0	115.1 ± 5.0
DEA	−500	200	700	400.0 ± 5.0	18.5 ± 3.0	5 ± 1	118.4 ± 5.0	89.6 ± 5.0
MDEA	−600	300	900	401.1 ± 5.0	29.0 ± 3.0	7 ± 1	140.6 ± 5.0	106.9 ± 5.0
AMP	−600	300	900	400.0 ± 5.0	25.4 ± 3.0	6 ± 1	155.3 ± 5.0	122.1 ± 5.0
PZ	−500	300	800	398.9 ± 5.0	36.0 ± 3.0	9 ± 1	151.2 ± 5.0	117.7 ± 5.0
Na_2_SO_3_	−600	300	900	400.0 ± 5.0	11.5 ± 2.0	3 ± 1	154.7 ± 5.0	118.7 ± 5.0
PolySep (PP)								
Initial	−300	200	500	436.9 ± 5.0	74.3 ± 3.0	17 ± 1	107.7 ± 5.0	83.2 ± 5.0
MEA	−400	200	600	437.0 ± 5.0	41.2 ± 3.0	9 ± 1	112.4 ± 5.0	84.7 ± 5.0
DEA	−500	300	800	441.1 ± 5.0	49.0 ± 3.0	11 ± 1	159.8 ± 5.0	125.6 ± 5.0
MDEA	−800	800	1600	439.0 ± 5.0	40.3 ± 3.0	9 ± 1	421.1 ± 5.0	335.0 ± 5.0
AMP	−390	300	690	442.3 ± 5.0	86.8 ± 3.0	20 ± 1	143.8 ± 5.0	112.1 ± 5.0
PZ	−500	500	1000	452.2 ± 5.0	43.4 ± 3.0	10 ± 1	186.2 ± 5.0	143.0 ± 5.0
Na_2_SO_3_	−180	100	280	443.6 ± 5.0	61.5 ± 3.0	14 ± 1	64.9 ± 3.5	51.4 ± 3.5
PES-020 (PES)								
Initial	−380	150	530	399.0 ± 5.0	14.3 ± 2.0	4 ± 1	62.8 ± 3.5	39.6 ± 3.5
MEA	−180	60	240	399.0 ± 5.0	31.8 ± 3.0	8 ± 1	45.1 ± 3.5	26.3 ± 3.5
DEA	−400	180	580	399.0 ± 5.0	31.5 ± 3.0	8 ± 1	101.6 ± 5.0	61.4 ± 5.0
MDEA	−300	150	450	400.0 ± 5.0	22.8 ± 3.0	6 ± 1	76.1 ± 5.0	46,0 ± 3.5
AMP	−170	50	220	399.0 ± 5.0	31.1 ± 3.0	8 ± 1	45.9 ± 3.5	27.9 ± 3.5
PZ *	−	-	-	-	-	-	-	-
Na_2_SO_3_	−600	200	800	400.0 ± 5.0	16.8 ± 2.0	4 ± 1	123.3 ± 5.0	73.9 ± 5.0
MCM-010 (Nylon PA)								
Initial	−400	400	800	399.0 ± 5.0	31.6 ± 3.0	8 ± 1	158.3 ± 5.0	128.9 ± 5.0
MEA *	−	-	-	-	-	-	-	-
DEA	−500	300	800	399.0 ± 5.0	30.9 ± 3.0	8 ± 1	152.4 ± 5.0	121.5 ± 5.0
MDEA	−600	700	1300	399.0 ± 5.0	47.6 ± 3.0	12 ± 1	214.8 ± 5.0	171.8 ± 5.0
AMP	−600	400	1000	399.0 ± 5.0	49.8 ± 3.0	12 ± 1	196.0 ± 5.0	159.7 ± 5.0
PZ	−390	200	590	400.0 ± 5.0	62.2 ± 3.0	16 ± 1	151.7 ± 5.0	122.7 ± 5.0
Na_2_SO_3_	−400	300	700	400.0 ± 5.0	67.8 ± 3.0	17 ± 1	165.5 ± 5.0	134.8 ± 5.0

* The samples were almost completely degraded into parts unsuitable for analysis after treatment.

**Table 5 membranes-13-00544-t005:** The estimation of the chemical and morphological stability of the porous polymeric membranes after exposure to degraded solvents, and a solution of an oxygen scavenger.

Membrane	Solvent																							
	MEA				DEA				MDEA				PZ				AMP				Na_2_SO_3_			
	I	II	III	Σ	I	II	III	Σ	I	II	III	Σ	I	II	III	Σ	I	II	III	Σ	I	II	III	Σ
PVDF-022 (PVDF)	−/+	+	−/+	+/−	+/−	+	+/−	−/+	+/−	+/−	+/−	+/−	−	−	−	−	−/+	−	−/+	−/+	+/−	+/−	−	−/+
MFF4-020 (PTFE)	+/−	+/−	+/−	+/−	+/−	−/+	+	+/−	+	+/−	+/−	+/−	+	+/−	+/−	+/−	+	+/−	+/−	+/−	+	−/+	+/−	+/−
PolySep (PP)	+/−	+/−	+	+/−	+/−	+/−	+/−	+/−	+/−	+/−	–	−/+	+/−	−/+	−/+	−/+	+/−	+/−	+/−	+/−	+/−	+/−	+/−	+/−
PES-020 (PES)	+/−	−	+/−	−/+	+/−	−	−/+	−/+	+/−	+/−	+/−	+/−	−	−	−	−	+/−	−	+/−	−/+	+/−	−/+	−/+	−/+
MCM-010 (Nylon PA)	−	−	−	−	−/+	+	+	+/−	−/+	+/−	+/−	−/+	+/−	−	+	−/+	+/−	+/−	+/−	+/−	−/+	−	+	−/+

I: estimation of chemical stability using FTIR spectra; II: estimation of morphological stability by changing porosity using AFM; III: estimation of morphological stability by changing roughness using AFM; Σ—total (average) estimation of membrane stability; +: the membrane is stable (changes within 0–10% from initial value for AFM data and no changes in FTIR spectra); +/−, light orange: the membrane is stable but there are slight changes (changes within 10–50% of the initial value for AFM data and mild changes in FTIR spectra); −/+, orange: the membrane is not stable (obvious changes within 50–100% of the initial value for AFM data and obvious changes in FTIR spectra are present); −, red: the membrane is not stable (the membrane is destroyed).

**Table 6 membranes-13-00544-t006:** Water contact angle before and after exposure to MEA.

Membrane	Initial, °	After Exposure to MEA, °
PVDF-022 (PVDF)	98.9 ± 2.0	86.9 ± 2.0
MFF4-020 (PTFE)	111.9 ± 2.0	110.9 ± 2.0
PolySep (PP)	100.3 ± 2.0	102.2 ± 2.0
PES-020 (PES)	76.1 ± 2.0	68.1 ± 2.0
MCM-010 (Nylon PA)	45.2 ± 2.0	33.8 ± 2.0

**Table 7 membranes-13-00544-t007:** Pore size of the MFF4-020 (PTFE) membranes before and after exposure to degraded solvents.

Solution	d_min_, μm	d_MFP_, μm	d_max_, μm
Initial	0.27 ± 0.01	0.43 ± 0.01	0.56 ± 0.01
MEA	0.22 ± 0.01	0.43 ± 0.01	0.95 ± 0.01
DEA	0.32 ± 0.01	0.43 ± 0.01	0.53 ± 0.01
MDEA	0.32 ± 0.01	0.43 ± 0.01	0.53 ± 0.01
AMP	0.36 ± 0.01	0.45 ± 0.01	0.53 ± 0.01
PZ	0.34 ± 0.01	0.44 ± 0.01	0.54 ± 0.01

## Data Availability

The data presented in this study are available on request from the corresponding author.

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
