# Peer review of "Stability of Porous Polymeric Membranes in Amine Solvents for Membrane Contactor Applications"

_membranes, 2023, doi:10.3390/membranes13060544_

Round 1
Reviewer 1 Report (Previous Reviewer 1)
Below my comments:
1. In the title, what do the authors mean with “under amine CO2 solvents conditions”? I suggest to review the English syntax of this sentence.
2. In the opening sentence of the abstract, the authors refer to the use of membrane contactors as possible solution for solvent degradation. However, in their revision of the manuscript in response to my comment 1 and 2, they claim that they do not intent to use MC only for O2 removal, but also for other applications in the amine process, as for example the desorption stage. This makes me very confused: what is the real target of their application? If it is the CO2 desorption stage, then the whole introduction should be changed. If it is O2 removal from CO2 saturated amines, then the temperature chosen for the experiments (100 C) is not appropriate. For both cases, they should look at CO2 saturated with amines: in fact, for both the desorption stage and the O2 removal concept, the membranes would be in contact with CO2-loaded solvents and their properties would change consistently (e.g., viscosity, surface tension and contact angle, reactivity). This was already mentioned in my previous review and the justification given (only 0.2 molCO2/mol amine remaining in MEA at 100C) is quite weak, considering that it refers to only 1 of the solvents they tested and that in the desorber the initial CO2 concentration will correspond to the highest loading exiting the absorber.
3. reporting the water contact angle to define the wettability of a solvent on a porous membrane has basically no use. What is important is the contact angle of the solvent on the membrane of interest. Without this assessment, the rest of the work done in this manuscript might be useless: if the solvent has a contact angle close or below 90 degrees, then the porous membrane cannot be used for the application and should be automatically discharged without further investigations.
4. the results reported in Table 7 are confusing: the average pore size remains basically unchanged even though the dmin or the dmax change quite a lot. What is this related to? Could it be related just to the accuracy of the measurement? When it comes to gas permeance, I suppose that the average pore size is the one that affects the transport properties, as the dmin or dmax could be related just to one specific pore. Therefore, this table does not really explain why the changes in gas permeance is rather limited even though the porosity reported in table 4 changes quite a lot upon exposure to solvents. The reasons behind the changes in porosity should also be discussed in the paper to strengthen the experimental observation.
5. the English grammar and syntax still require a thorough revision.
Author Response
Dear Dr. Magdalena Kowalska and Reviewer, thank you very much for your valuable comments, which allowed us to improve our manuscript. Please, find the replies to the Reviewer comments below.
- In the title, what do the authors mean with “under amine CO2 solvents conditions”? I suggest to review the English syntax of this sentence.
The title has been corrected.
- In the opening sentence of the abstract, the authors refer to the use of membrane contactors as possible solution for solvent degradation. However, in their revision of the manuscript in response to my comment 1 and 2, they claim that they do not intent to use MC only for O2 removal, but also for other applications in the amine process, as for example the desorption stage. This makes me very confused: what is the real target of their application? If it is the CO2 desorption stage, then the whole introduction should be changed. If it is O2 removal from CO2 saturated amines, then the temperature chosen for the experiments (100 C) is not appropriate. For both cases, they should look at CO2 saturated with amines: in fact, for both the desorption stage and the O2 removal concept, the membranes would be in contact with CO2-loaded solvents and their properties would change consistently (e.g., viscosity, surface tension and contact angle, reactivity). This was already mentioned in my previous review and the justification given (only 0.2 molCO2/mol amine remaining in MEA at 100C) is quite weak, considering that it refers to only 1 of the solvents they tested and that in the desorber the initial CO2 concentration will correspond to the highest loading exiting the absorber.
Thank you for your comment. The initial part of the abstract and introduction section has been significantly corrected with the focus on stability of membranes within different MC applications in amine absorption technology (absorption, CO2 desorption, deoxygenation, lines 37-68) according to your proposal. We believe that the corrected text will be more clear for the Reader.
We definitely agree with the Reviewer that CO2 will greatly affect the properties of amine solvents. And we will use the carbonated solvents in our future studies of the stability of membranes. However, in this work,, we tried to use pure amine solvents with model degradation products with the exception of a very complex CO2 influence at the first stage of our project.
- reporting the water contact angle to define the wettability of a solvent on a porous membrane has basically no use. What is important is the contact angle of the solvent on the membrane of interest. Without this assessment, the rest of the work done in this manuscript might be useless: if the solvent has a contact angle close or below 90 degrees, then the porous membrane cannot be used for the application and should be automatically discharged without further investigations.
Thank you for your comment. In the studies performed, the measurement of the water contact angle before and after exposure to amine solvents is a fast way to determine changes in the surface properties of membranes.
We agree with the comment of the Reviewer that the perspective of the application of the particular membrane in MC is mainly driven by its wettability with amine solvent, which can be represented by solvents contact angle measurements. We also agree with the comment of the Reviewer that the hydrophobic membranes should be used for this application.
However, in our case, the studied materials are of special interest for use as supports for composite membranes. Theoretically, in this case, slightly less hydrophobic materials (contact angle ≤90°) can also be used due to the fact that all direct negative effects of amine are prevented by a thin hydrophobic nonporous selective layer. However, the penetration of solvent vapors is expected from the evident experiment data. See, for example, doi: 10.1016/j.ijggc.2015.07.032. This can inevitably affect the porous support after solvent vapor sorption and condensation in the porous structure of the support membrane. Thus, the surface properties of the support membrane can be change. Keeping this in mind, we did water contact angle measurements to figure out the influence of amine media to surface properties of membranes.
- the results reported in Table 7 are confusing: the average pore size remains basically unchanged even though the dmin or the dmax change quite a lot. What is this related to? Could it be related just to the accuracy of the measurement? When it comes to gas permeance, I suppose that the average pore size is the one that affects the transport properties, as the dmin or dmax could be related just to one specific pore. Therefore, this table does not really explain why the changes in gas permeance are rather limited, even though the porosity reported in Table 4 changes quite a lot upon exposure to solvents. The reasons behind the changes in porosity should also be discussed in the paper to strengthen the experimental observation.
Table 7 shows the average values obtained from several measurements. Regarding the influence on the transport characteristics of membranes, we can say that it is true, you are right, the average porosity determines the gas permeance of the membrane. In this case, the surface porosity determined by AFM, in fact, may not be directly related to the transport characteristics of membranes. It is most important in terms of the potential use of the membrane as a support for a composite membrane. Indeed, changes in the permeance of the samples are associated with a complex change in the surface porosity and porous structure of the membranes. The average porosity and pore size mainly determine the gas permeance of the membranes. Thus, in the case of PTFE membranes, the gas permeance changed slightly. This phenomenon is a result of minor variation of transport porosity (dMFP remains practically unchanged after exposure to amine solvents, Table 7). It should be noted that the surface porosity presented in Table 4 varies more significantly (20-60%). In our point of view, this seemingly unexpected result is due to the fact that surface porosity may not be directly related to the transport porosity of membranes. As stated, mechanical destruction of biaxially oriented PTFE fibrils results in surface ‘etching’ and significant changes of surface porosity but mild changes of gas transport properties. In accordance with your important remark, we have tried to reflect arguments about influence of porous structure in more detail and add these arguments to the discussion of the results (lines 421-439).
- the English grammar and syntax still require a thorough revision.
This was taken care of.
Reviewer 2 Report (Previous Reviewer 2)
One remaining question I have for the paper is that the porosity and pore size change of PTFE are different from all other polymers in the list. I am not sure if that can be an excellent example to present in the paper and draw the conclusions that authors had. In detail, PTFE is the only membrane showing a trend of smaller porosity after impurity exposure, and all others` porosity is relatively increased. What are the permeance and selectivity of the rest of the membranes in the list? Please show them in the manuscript to support your conclusion.
One remaining question I have for the paper is that the porosity and pore size change of PTFE are different from all other polymers in the list. I am not sure if that can be an excellent example to present in the paper and draw the conclusions that authors had. In detail, PTFE is the only membrane showing a trend of smaller porosity after impurity exposure, and all others` porosity is relatively increased. What are the permeance and selectivity of the rest of the membranes in the list? Please show them in the manuscript to support your conclusion.
Author Response
Dear Dr. Magdalena Kowalska and Reviewer, thank you very much for your valuable comments, which allowed us to improve our manuscript. Please, find the replies to the Reviewer comments below.
1) One remaining question I have for the paper is that the porosity and pore size change of PTFE are different from all other polymers in the list. I am not sure if that can be an excellent example to present in the paper and draw the conclusions that authors had. In detail, PTFE is the only membrane showing a trend of smaller porosity after impurity exposure, and all others` porosity is relatively increased. What are the permeance and selectivity of the rest of the membranes in the list? Please show them in the manuscript to support your conclusion.
Thank you very much for your comment. Data on the properties of all non-destroyed membranes are presented in Supplementary Materials so that the text does not unnecessarily increase. However, in accordance with your comment, more emphasis on this is added to the text with an example on a different type of membrane (p. 13, lines 433-439).
Reviewer 3 Report (Previous Reviewer 4)
Authors have improved certainly their manuscript which now reads more scientific and clearer. Their answers to my queries are, mostly, reasonable and well-based, and they have clarified appropriately the comments I suggested. Still some comments need to be answered before acceptance:
-Table 1: usually manufacturer information is termed as nominal values, not declared ones. ON the other side there is an strange * at the bottom of the table. Authors claim “pore size was determined using the liquid-liquid displacement porosimetry”. What pore sizes, those included as declared ones? Normally nominal values are measured by companies but they do not declare how they arrived to such values. And it should be strange that different companies measuring the pores with the same technique. Or, perhaps authors reefer to values measured by them? Then this call is not properly placed in this table. Confusing
- Figure 1: I wonder if final aspect of this figure and the FYIR spectra will be that shown in the copy for reviewing. It has low definition, change it
- Table 5: which is the meaning of the coloured values in some of the data? Explain properly in text. ON the other side, if some information says not stable and other one says stable, which is the total estimation of stability? Average?
- Table 7: surely Porometer equipment gives you the resulting values in nm, but attending to the size of the pores in these supports (totally in the MF range) it is more common to express pore sizes in microns
Author Response
Dear Dr. Magdalena Kowalska and Reviewer, thank you very much for your valuable comments, which allowed us to improve our manuscript. Please, find the replies to the Reviewer comments below.
-Table 1: usually manufacturer information is termed as nominal values, not declared ones. ON the other side there is an strange * at the bottom of the table. Authors claim “pore size was determined using the liquid-liquid displacement porosimetry”. What pore sizes, those included as declared ones? Normally nominal values are measured by companies but they do not declare how they arrived to such values. And it should be strange that different companies measuring the pores with the same technique. Or, perhaps authors reefer to values measured by them? Then this call is not properly placed in this table. Confusing
Error corrected, footnote removed. This phrase was erroneously left in the correction at the previous stage of work on the manuscript and removed in this vertion.
- Figure 1: I wonder if final aspect of this figure and the FYIR spectra will be that shown in the copy for reviewing. It has low definition, change it
The definition of the Figure 1 has been increased and the Figure has been changed
- Table 5: which is the meaning of the coloured values in some of the data? Explain properly in text. ON the other side, if some information says not stable and other one says stable, which is the total estimation of stability? Average?
The captions to Table 5 have been clarified for better understanding by readers. Yes, that's right, the total estimation of stability was carried out by averaging.
- Table 7: surely Porometer equipment gives you the resulting values in nm, but attending to the size of the pores in these supports (totally in the MF range) it is more common to express pore sizes in microns
This has been corrected.
This manuscript is a resubmission of an earlier submission. The following is a list of the peer review reports and author responses from that submission.
Round 1
Reviewer 1 Report
The submitted manuscript investigates the compatibility of porous membrane and different amine-based solvents for the application of membrane contactors for the removal of O2 in amine absorbents, aiming at reducing oxidative degradation. The authors exposed different porous membranes (PVDF, PP, PTFE, PA and PES) to different solvents that can potentially be used in absorption-based CO2 capture process, investigating structural and morphological changes upon exposure. Although the subject is worth of investigation, I have the following concerns/comments about the presented research work:
1. in several part of the manuscript, the authors mention that composite membranes would be the most appropriate solution for O2 removal from amine absorbents, but then they investigate only porous membranes. Although these porous membranes can potentially be used as support for composite membranes, it is unclear why compatibility should be investigated. In the case of composite membranes, only the thin dense layer will be exposed to the amine absorbents, therefore preventing any possible effect on the porous structure beneath. This makes the manuscript not really interesting for the final application. The authors should expand the research including potential dense polymers that can be potentially used as coating for thin film composite membranes.
2. The choice of the testing temperature (100 C), makes also the purpose of the study quite unclear. If the target is to use membrane contactor (MC) for O2 removal, then the MC will be placed right after the absorber, where a temperature around 60 C can be expected. Why did the authors choose 100 C? A good explanation must be provided.
3. Another important aspect is related to the fact that if the target is O2 removal from a loaded solvent, then the compatibility tests should be performed with loaded solvents. When amines are chemically bonded with CO2, their reactivity is significantly reduced. In the experimental procedure design by the authors, no CO2 loading is mentioned, and it can be assumed that unloaded amines are used. Once again, the tested conditions are not representative for the targeted application.
4. When it comes to compatibility between membranes and amine absorbent, wetting is a very important aspect to consider. In case of wetting, the membrane should be discarded, without further investigation. However, the authors do not report any information about wetting of the used solvents with respect to the considered membranes. Contact angle analysis can be performed to confirm that wetting is not an issue, but typically this is a measurement done only at room temperature. For higher temperatures, it is possible to investigate the uptake of the liquid in the porous layer (if wetting happens, then the liquid will fill the pores). Some consideration around this must be reported in the paper to confirm that the chosen polymeric materials can be used in the target application.
5. The English grammar and syntax should be carefully revised as several errors are present in the text, making sentences difficult to fully understand.
6. From the introduction, it appears as the authors have proposed the use of membrane contactors for removal of O2 from amine absorbents for the first time. However, this is something that TNO has worked at least since 2018 (they also own several patents about this), as it is also mentioned in one of their latest publications (Figueiredo et al., Impact of dissolved oxygen removal on solvent degradation for post-combustion CO2 capture, 2021, doi: 10.1016/j.ijggc.2021.103493). For completeness, this should be included in the introduction.
7. In the introduction, the authors cite several references (from 24 to 40), where many are quite dated and not focusing on CO2 capture. Compatibility between porous and non-porous membrane contactors with amine absorbents have been largely investigated in the past decade and the authors should give a better overview of this in the introduction. In addition, some of these references can also be used for comparison of the results obtained in this work (no literature comparison is reported). For example, the paper from Ansaloni et al. (Ind. Eng. Chem. Res. 2016, 55, 51, 13102–13113) reports on the compatibility between PP and PTFE with 30% MEA and also loaded solvents.
8. page 6, line 220: PVDF is known to be a hydrophobic polymer, but the authors report that at least 10 mol% water is detected in the porous structure with FTIR. This is quite unexpected given the nature of the polymer. How do the authors explain such observation? Can be they support their funding with literature evidence?
9. in table 5, there is a very large inconsistency between the compatibility results obtained with the 2 methods. This makes the conclusions very confusing. How do the authors explain such large inconsistency? The table should be presented in a more understandable manner.
10. page 11, line 331: what is a “violation of the surface morphology”?
11. page 11, line 335: the conclusions reported here about PTFE are not supported by Table 5 (PTFE can only be used with MEA and DEA, which are not going to be absorbents used in real processes).
12. Figure 3: according to table 4, porosity of PTFE dropped from 12 to 5 in case of DEA, but a minimum reduction is observed from the measurement of the N2 permeance. Such large drop in porosity (even if just on the surface) should have a larger impact on the gas permeance. In general, there is a very poor correlation between porosity in table 4 and the measured gas permeance. How do the authors explain these results?
Reviewer 2 Report
The authors studied the effect of the impurities in the amine solution on different commercial membranes. The morphology changed when membranes exposure to different environments.
Minor changes are needed before it publishes.
- Can the author provide a deep analysis of why the different impurities affect porosity and roughness?
- Is there any correlation between porosity and separation performance before and after the amine solution exposure?
Reviewer 3 Report
Review of the paper
“Stability of porous polymeric membranes for membrane contactors under amine CO2 solvents conditions”
By: Denis Kalmykov, Sergey Shirokikh, Evgenia A. Grushevenko, Sergey A. Legkov, Galina N. Bondarenko, Tatyana S. Anokhina, Sergey Molchanov and Stepan D. Bazhenov
Journal: Membranes.
Manuscript ID: membranes-2335114.
The paper presents study of chemical and morphological stability of different polymeric membranes to the prolonged exposure to various types of alkanolamines with the addition of heat stable salts anions typically used as oxygen scavengers. The topic is of interest for the membrane field.
Five polymers are considered to that aim: PVDF, PTFE, PP, PES and PA(nylon), and five types of amines, with inappropriate emphasis considered as “a wide range” (see line 143). The effects of the standard treatment considered were analyzed through FTIR spectroscopy and atomic force microscopy (AFM), pointing out the variations in bond spectra and changes in thickness, porosity and roughness. Conclusions about the chemical and morphological stability are summarized in Table 5, considering separately the FTIR and AFM results. Overall, all the polymers examined were considered not chemically and morphologically stable with respect to the treatment considered, with the exception of PTFE. In fact, the results found would require a more detailed discussion insofar as some polymers show stability with respect to some amines even not to all, and for some amines PTFE shows good chemical stability (positive response to FTIR analysis), but not a good response to chemical stability (negative response to AFM analysis). That should be discussed in more detail.
Overall the manuscript can be improved by a careful revision to eliminate several language errors or improper writings as well as the lack of care in several places (e.g. in Eqs. (1) and (2) the quantities Ra and Rb are defined, while in lines 201 and 202 they are indicated as Sa and Sb; line 305 mentions RA membrane samples which should likely be PA membrane samples.
In addition, the data reported in Table 4 should all be given with their respective errors, otherwise the appreciation of the changes is lost; the indications of errors are also required for the permeance and selectivity data shown in Figs. 3 and 4, respectively.
In conclusion:
The paper presents data which may be of interest and can be published only after a major revision according to the comments above.

Reviewer 4 Report
Authors of the present work aimed to study the stability of several commercial porous polymeric membranes to the exposure to various types of alkanolamines with the addition of heat stable salts anions. Certainly this is a sort of information that is normally issued by the membrane manufacturers in their brochures. Anyway, as concerned with some very specific conditions, this study could have some interest. Nevertheless, as authors claim, their study is focussed on the damage experienced by some porous supports, usually intended for making composite membrane contactors. In that case, the performance properties of the resulting contactors and, especially the selectivity of them, or the loss of selectivity due to solvent damage, will be more related with the active layer of the composite, than with the support. Certainly those polymers that resulted absolutely damaged in the conditions of the study, are not suitable for such membrane composites. But even those not severely damaged could be useful after appropriate active layer addition.
On the other side, I think the techniques used for membrane characterization of the supports are not the most useful ones. FTIR reflects more the presence or absence of certain chemical groups but not too much about how these groups added during exposure to different solvents can damage membrane structure.
Similarly, AFM could be not the best idea for testing changes in the membranes. Certainly some of the figures (only one is shown in the paper and the rest in the supplementary material, I think including some AFM micrographs more for different conditions of damage could be interesting) show different degrees of structural changes. But the best way to assure this damage is the use of some porosimetric technique, simply analysing the AFM pictures with an image analysis software could be enough to get an idea about pore size changes. Even better to use some porometry (GLDP for example) to check how important are these changes. Certainly the selectivity values measured for the supports are not very useful as it should be controlled by the deposited active layer.
In my opinion the paper has not been adequately written, presenting a poor English and many incorrect or unclear phrasing. Authors have put on a lot of experimental work but they have failed on explaining properly their results.
Some other minor comments follow:
- Lines 54-56: rewrite, bad grammar and unclear
- Line 64: at the same time makes no sense here. Similarly in line 96. Change by “on the other side”
- Line 122: not accordingly, high temperature is the cause, not the consequence
- Section 2.1.1: five materials really, In any case, I wonder if the manufacturers have not other membranes from same material. Usually commercial membranes are characterized by their trading name and pore size/cut-off. Finally the selection of these membrane supports and no other was based in the fact they were previously characterized by [45] or there is another reason for that.
- Lin 149: normally region, street and zip code of the company is not necessary, just the city and country and once described location first time it is not needed to comment everywhere (lines 153-154)
- Line 170: I wonder if you´re sealing was not damaged after 14 days at 100 ºC
- Line 179: frustrated ATR? Did it not work?
- Line 187: what happened with the destroyed samples, not stated anywhere which membrane or which conditions leaded to destruction
- Line 196: mask? Rewrite all paragraph, absolutely unclear
- Section 2.2: you should state somewhere that characterization studies were conducted on virgin supports and then, on solvents exposed ones.
- Figure 4: what means ideal selectivity, real or not?